# Integrated Analyses of Single-Cell Transcriptome and Mendelian Randomization Reveal the Protective Role of Resistin in Sepsis Survival in Intensive Care Unit

**DOI:** 10.3390/ijms241914982

**Published:** 2023-10-07

**Authors:** Hanghang Chen, Haihua Luo, Tian Tian, Shan Li, Yong Jiang

**Affiliations:** Guangdong Provincial Key Laboratory of Proteomics, State Key Laboratory of Organ Failure Research, Department of Pathophysiology, School of Basic Medical Sciences, Southern Medical University, Guangzhou 510515, China; hanghang461@i.smu.edu.cn (H.C.); btxlhh@smu.edu.cn (H.L.); tiantian321@i.smu.edu.cn (T.T.); lishan19900325@i.smu.edu.cn (S.L.)

**Keywords:** sepsis, resistin, genome-wide association study (GWAS), quantitative trait loci (QTL), Mendelian randomization (MR)

## Abstract

The high morbidity and mortality rates associated with sepsis highlight the challenges of finding specific remedies for this condition in the intensive care unit (ICU). This study aimed to explore the differentially expressed genes (DEGs) specific to cell types in sepsis and investigate the role of resistin in the survival of sepsis patients through Mendelian randomization (MR) analyses. We used single-cell and bulk transcriptome data to identify cell type-specific DEGs between sepsis and healthy controls. MR analyses were then conducted to investigate the causal relationships between resistin (one of the identified DEGs) levels and the survival of sepsis patients. Additionally, we utilized meQTL (methylation quantitative trait loci) to identify cytosine-phosphate-guanine (CpG) sites that may directly affect sepsis. We identified 560 cell type-specific DEGs between sepsis and healthy controls. Notably, we observed the upregulation of resistin levels in macrophages during sepsis. In bulk transcriptome, RETN is also upregulated in sepsis samples compared with healthy controls. MR analyses revealed a negative association existed between the expression of resistin, at both gene and protein levels, and the mortality or severity of sepsis patients in ICU. Moreover, there were no associations observed between resistin levels and death or organ failure due to other causes. We also identified three methylation CpG sites, located in RETN or its promoter region—cg06633066, cg22322184, and cg02346997—that directly affected both resistin protein levels and sepsis death in the ICU. Our findings suggest that resistin may provide feasible protection for sepsis patients, particularly those with severe cases, without serious side effects. Therefore, resistin could be a potential drug candidate for sepsis treatment. Additionally, we identified two CpG sites, cg06633066 and cg22322184, that were associated with RETN protein levels and sepsis death, providing novel insights into the underlying mechanisms of sepsis.

## 1. Introduction

Sepsis is a major cause of disease burden and mortality worldwide. With approximately 49 million sepsis cases and 11 million deaths globally, sepsis-related fatalities account for approximately 20% of all deaths [1]. Given the complexity and heterogeneity resulting from various routes of infection, specific therapies for sepsis are currently lacking.

Resistin was initially discovered and named for its role in insulin resistance in 2001 [2]. It has since been found to play important roles in metabolism, inflammation, and autoimmune diseases [3]. In contrast to its role in insulin resistance or diabetes, resistin may play a more critical role in sepsis [4]. Serum levels of resistin have been found to be significantly elevated in sepsis patients compared to healthy controls or non-sepsis patients, and this elevation was not found to be associated with pre-existing type 2 diabetes or obesity [4]. An observational study also found that resistin levels were higher in infants with severe sepsis or those who required mechanical ventilation [4]. In an observational study, resistin was higher among infants with severe sepsis or those who needed mechanical ventilation [5], indicating the potential for resistin in the diagnosis and treatment of sepsis. Nevertheless, the associations between resistin and sepsis remain to be fully understood.

Mendelian randomization (MR) is an approach that, like randomized controlled trials (RCTs), can provide a means to overcome potential limitations of observational studies and determine potential causal relationships between molecular traits and phenotypes [6]. Compared to RCTs, MR can offer a less expensive, faster, and more ethical assessment of the long-term impacts of exposures on outcomes [7]. The joint analysis of multiple quantitative trait loci (QTLs) and genome-wide association studies (GWASs) can facilitate the identification of variants associated with both disease risk and molecular traits [8]. For example, Huang et al. used MR to determine the causal relationship between the inhibition of 3-hydroxy-3-methylglutaryl-CoA reductase (HMGCR), the target of statins and a key enzyme in cholesterol biosynthesis, and the reduced risk of COVID-19 hospitalization [9].

This study aimed to estimate the association between resistin and sepsis by integrating multiple QTLs (expression QTLs, protein QTLs, and metabolite QTLs) with GWAS summary data. Notably, we provided the first demonstration of a specific causal and protective role of resistin in sepsis-related mortality among ICU patients. Furthermore, we investigated the underlying CpG sites located within RETN or its promoter, which may affect sepsis outcomes. Our analysis identified two CpG sites, cg06633066 and cg22322184, that were found to be associated with both RETN protein levels and sepsis-related mortality. In summary, our findings suggest that resistin may represent a potential drug for sepsis treatment, particularly among critical patients.

## 2. Results

The schematic diagram is displayed in Figure 1A. The overall study design is illustrated in Figure 1B.

### 2.1. Identification of 560 DEGs between Healthy Controls and Sepsis Samples

The averaged expression levels and ratios of cell markers that were used for cell annotation were displayed as a dot plot (Appendix A). All cells were classified into ten types and displayed in the UMAP (Figure 2A). Moreover, we found 560 cell type-specific differentially expressed genes (DEGs) in six cell types between healthy controls and sepsis samples (Figure 2B). Notably, we observed that RETN was upregulated in macrophages during sepsis compared with healthy controls. Therefore, we further investigated the causal relationship between RETN levels and sepsis. 

### 2.2. RETN Was also Upregulated in Sepsis Samples in Bulk Transcriptome

The expression levels of RETN were upregulated in sepsis samples compared to healthy controls in bulk transcriptome data (Figure 2D). The expression levels of RETN were upregulated in samples of deceased patients compared to controls of survivors (Figure 2E). The heatmap displays the normalized expression levels of RETN in each sample. All samples were arranged as per their groups (sepsis and healthy), in hospital mortality (deceased and survived), and the Sequential Organ Failure Assessment (SOFA) scores.

### 2.3. The Associations between RETN and Outcomes

We employed LD-clumping to remove QTLs with high linkage disequilibrium (LD) in RETN eQTLs and pQTLs. The resulting independent eQTLs and pQTLs were used as instrumental variables (IVs) in our analysis. The lead eQTL (rs149007767) and the lead pQTL (rs3745368) were identified based on their respective smallest *p* values.

We generated Manhattan plots for the independent eQTLs (Figure 3A) and pQTLs (Figure 3B) using the “CMplot” R package [10]. The *p* values for three pQTLs (rs3745368, rs117029024, and rs149110519) were extremely low (6.75 × 10^−192^, 1.21 × 10^−122^, and 4.34 × 10^−68^, respectively), and we compressed these values to 1 e−50 for better display of other pQTLs in the Manhattan plot. The Z scores of eQTLs (Appendix A) and pQTLs (Appendix A) were also presented. All statistical values related to RETN eQTLs and pQTLs are listed in Appendix A.

Our analysis revealed that RETN levels were negatively associated with sepsis-related 28-day mortality in the ICU. The eQTL data supported the association between RETN RNA levels and a lower risk of sepsis-related death, while the cis-eQTL data supported this association as well as the association between RETN RNA levels and a lower risk of sepsis severity. The pQTL data supported the association between resistin protein levels and a lower risk of sepsis-related death and severity, while the cis-pQTL data supported these associations as well. These negative associations between all exposures and outcomes related to sepsis were presented in Figure 4, and a forest plot summarizing the results was shown in Figure 5. Additionally, we found that the eQTL data supported the association between RETN RNA levels and a lower risk of pneumonia-related death, while the cis-eQTL, pQTL, and cis-pQTL data did not support this association. The sensitivity analysis results were listed in Appendix A.

Our analysis did not find any significant association between RETN levels and other secondary outcomes such as pneumonia severity, pneumonia death in ICU, severe COVID-19 infection with respiratory failure, heart failure, and so on. The results of MR analyses between RETN (eQTL and pQTL) and all outcomes are presented in Table 1.

All associations between eQTL, pQTL, and main outcomes of sepsis passed the sensitivity and directionality tests.

### 2.4. The MR Results of RETN-Related CpG Sites and RETN Levels and Outcomes of Sepsis

RETN-related CpG sites were defined as CpG sites that were found to have associated SNPs within the RETN gene or promoter region, with a *p*-value of less than 5 × 10^−8^. We identified a total of 12 CpG sites that met this criterion. Our MR analysis revealed that four CpG sites, cg15460739, cg15828235, cg17474222, and cg24433207, were positively associated with RETN mRNA levels, while cg02383368 was found to be negatively associated with RETN mRNA levels. Three CpG sites, cg02346997, cg06633066, and cg22322184, were negatively associated with RETN protein levels, while cg17474222 was positively associated with both mRNA and protein levels of RETN.

Furthermore, three CpG sites, cg02346997, cg06633066, and cg22322184, were found to be associated with an improved risk of sepsis-related death. The MR results for all RETN-related CpG sites and RETN levels, as well as outcomes related to sepsis, are listed in Table 2.

### 2.5. Colocalization Analyses Show the Colocalization between Some CpG Sites and RETN Protein Levels

Interestingly, the three CpG sites, cg02346997, cg06633066, and cg22322184, were found to be associated with both RETN protein levels and increased risk of death in MR analyses. We conducted colocalization analyses, which showed that these CpG sites are colocalized with the risk of sepsis-related death via a shared SNP-rs3745367, located within the second intron of RETN. Results of all colocalization analyses are listed in Table 3. Furthermore, we conducted sensitivity analyses, including colocalization analyses that did not identify any shared causal variants, which support the robustness of Mendelian randomization findings.

## 3. Materials and Methods

### 3.1. Single-Cell Transcriptome Data

We collected single-cell transcriptome data from GEO database (https://www.ncbi.nlm.nih.gov/geo/query/acc.cgi?acc=GSE167363, accessed on 15 December 2022) [11]. This data contain peripheral blood mononuclear cells (PBMC) from two healthy donors and five sepsis patients. Further details regarding this dataset can be found in the original published paper [11].

### 3.2. Bulk Transcriptome Data

We downloaded bulk transcriptome data from GEO database (https://www.ncbi.nlm.nih.gov/geo/query/acc.cgi?acc=GSE185263, accessed on 15 December 2022) [12]. This data contain blood bulk transcriptome data of 348 sepsis patients and 44 healthy controls.

### 3.3. Exposure Data

All exposure and outcome cohorts were individuals of European descent. The eQTL data of RETN (gene symbol of resistin) were collected from eQTLGen Consortium (https://www.eqtlgen.org/cis-eqtls.html, accessed on 10 February 2023) [13]. This dataset includes information on 10,317 SNPs (single nucleotide polymorphisms) and expression levels of 19,942 genes for each of the 31,684 sequenced blood samples.

The pQTL data of resistin were collected from the deCODE database (https://www.decode.com/summarydata/, accessed on 22 February 2023) [14]. This dataset includes information on 18,084 SNPs and 4907 plasma proteins for each blood sample of 35,559 Icelanders, sequenced using SomaScan version 4.

The meQTL data were extracted from the GoDMC database (http://mqtldb.godmc.org.uk/downloads, accessed on 27 February 2023) [15]. This dataset includes information on 27,750 blood samples sequenced using Illumina 450 k. The cis-meQTL was defined as the meQTL within a 1 Mb distance from the methylation site and was publicly available. Additional details regarding these datasets can be found in the original publications.

### 3.4. Outcome Cohorts

The primary outcomes of this study were three sepsis-related outcomes, i.e., the incidence of sepsis, sepsis requiring critical care admission, and 28-day mortality in the intensive care unit (ICU) following an episode of sepsis. Summary-level data for the sepsis GWAS dataset are available in the UK Biobank. In the Integrative Epidemiology Unit (IEU) OpenGWAS project database (https://gwas.mrcieu.ac.uk/, accessed on 10 February 2023) [16], these outcomes were identified by the IDs ieu-b-4980, ieu-b-4982, and ieu-b-4981, respectively. The data are easily accessible as VCF files or through the “TwoSampleMR” R package.

The secondary cohorts in this study consisted of individuals who experienced death or organ failure due to various reasons. These cohorts were extracted from the IEU database. The aim of this analysis was twofold: firstly, to investigate the potential causal role of RETN or resistin in other severe diseases; and secondly, to demonstrate the safety of RETN or resistin without any serious side effects.

### 3.5. Identification of DEGs between Healthy Controls and Sepsis Samples

We followed the “Seurat” R package [17] pipeline for our single-cell RNA-seq data analysis. We removed unqualified cells based on specific criteria (gene counts per cell ≤300, gene counts per cell ≥30,000, percent of mitochondrial genes per cell ≥20%, and percent of hemoglobinic genes per cell ≥3%). The expression matrix of all cells was then normalized using the “LogNormalize” method. Next, we identified 2000 highly variable genes among the cells using the “vst” method. To reduce the dimensions of the data, we conducted principal component analysis (PCA) and selected the top 30 principal components to cluster cells based on the marker genes, as determined with an elbow plot. We constructed a shared nearest neighbor graph by calculating the neighborhood overlap between each cell and its nearest neighbors. Finally, we identified cell clusters using a shared nearest neighbor (SNN) modularity optimization-based clustering algorithm.

We utilized the uniform manifold approximation and projection (UMAP) to visualize a landscape of all cells, and annotated cell clusters using cell markers such as CD3 for T cells. Subsequently, we identified six clusters of differentially expressed genes (DEGs) between normal and tumor cells in six cell types (|log fold change (logFC)| > 1 and *p*-value < 0.05). We excluded several cell types, including neutrophils that only appeared in sepsis, during the calculation of DEGs.

### 3.6. Exploring RETN Differential Expression in Bulk Transcriptome

We utilized the “DESeq2“ R package pipeline for our bulk RNA-seq data analysis [18]. We transformed and normalized the data using “DESeqDataSetFromMatrix” and “vst” functions.

### 3.7. The Association between RETN Expression and Outcomes

As RETN was found to be upregulated in macrophages during sepsis, we conducted further investigation into the potential roles of RETN in sepsis using the MR approach. The MR approach is based on three key assumptions: (i) the instrumental variables (IVs) are associated with the exposure, (ii) there are no confounders of the associations between the IVs and the outcome, and (iii) the IVs are associated with the outcome solely through the exposure [19].

The inverse variance weighted (IVW) method is considered the most powerful and commonly used MR method [20,21]. In our study, we employed the IVW method and utilized the “TwoSampleMR” R package to conduct our two-sample MR analyses. In cases where only one SNP was available, the Wald ratio method was used as the sole option [22].

To begin our MR analysis, we began by searching for instrumental variables (IVs) for the exposure. IVs were defined as SNPs that are correlated with the exposure (with a *p*-value < 5 × 10^−8^). We then conducted LD clumping for the IVs to identify independent SNPs. SNPs with linkage disequilibrium (LD) (defined as r2 < 0.01 within clumping distance 10,000 kb) were excluded. The F value of each instrumental variable was calculated using the formula: F = (beta/se)^2^ (where se represents the standard error) [16]. To avoid weak instrumental variable bias, instrumental variables with an F value < 10 were removed [22]. To represent results for diverse traits on a common scale, we introduced a Z score. The Z score represents the scaled direction and efficiency of the association between the SNP and the trait [23]. The Z score was calculated using the formula: Z score = beta/se [16].

Next, we extracted the same SNPs in the outcome data. If some SNPs could not be found in the outcome data, then we chose not to find proxies. The minor allele frequency (MAF) threshold was defined as 0.01 [24].

Finally, effect alleles of SNPs between the exposure and outcome were harmonized to be relative to the same allele. MR results could be obtained using the “mr” function on the harmonized data. The odd ratio (OR) could be calculated using the formula: OR = exp(beta) [16].

In addition to analyzing the potential role of RETN in sepsis using the MR approach, we also conducted an evaluation of the causal effect of RETN cis-eQTL on the outcomes of interest. Cis-eQTL was defined as the SNPs with a distance of 1 Mb from RETN [25]. Due to the nature of cis-eQTL analysis, the stringent parameters used in the previous analyses were not suitable. Therefore, linkage disequilibrium (LD) was loosely defined as r2 < 0.3 within clumping distance of 100 kb [26]. The follow-up analyses for cis-eQTL were the same as those for eQTL.

### 3.8. The Association between RETN Protein and Outcomes

RETN pQTL was clumped using 1000 genomes phase 1 v3 reference panel of EUR and “PLINK” software (Version: 1.90) [27] to remove the LD [28]. The follow-up analyses were the same as analyses of eQTL.

### 3.9. The Associations between cis-meQTL and RETN eQTL, pQTL, and Outcomes of Sepsis

To explore the potential influence of methylation on the expression of RETN and its association with sepsis outcomes, we identified CpG sites that directly influence RETN expression, protein levels, or sepsis outcomes. As most cytosine-phosphate-guanine (CpG) sites are located within the promoter region [29], we chose to focus on SNPs within the promoter region (2 kb upstream sites) or gene region of RETN as target instrumental variables (IVs). We searched for CpG sites that were correlated with these SNPs (with a *p*-value < 5 × 10^−8^) and defined them as RETN-related CpG sites. We then investigated the causal functions of these CpG sites on RETN eQTL, pQTL, and outcomes related to sepsis.

### 3.10. Sensitivity Test

We used Cochran’s Q statistics [30] to quantify the heterogeneity and MR-Egger method [31] to test the pleiotropy of the harmonized data. Q or *p* value > 0.05 was defined without heterogeneity or pleiotropy. Additionally, the MR Steiger directionality test was used to exclude the potential reverse causalities [32].

### 3.11. The Colocation Analyses

To investigate the potential influence of the same causal variant(s) on multiple traits, we performed colocation analyses using the “coloc” R package [33], which is based on the Bayesian method. We examined the colocalization of CpG sites, RETN, and the two outcomes of sepsis in the promoter (2 kb upstream sites) or gene region of RETN (chr19:7,731,935–7,735,341 on hg19 by Ensembl). We calculated the Varbeta value of each instrument variant using this formula: varbeta = se^2^, and the Z-score was determined using this formula: z-score = beta/se. We then used the H1 posterior probability (PP.H1), PP.H2, PP.H3, and PP.H4 to assess the association of each variant with trait 1 and/or trait 2. We interpreted PP.H4 > 0.8 [34], indicating colocation between two traits through at least one shared SNP.

### 3.12. Statistical Methods

The expression levels between two groups were compared using “Wilcoxon” method. MR analyses were performed using IVW methods if there were more than one shared SNPs in exposure and outcome data. Wald ratio method was used if there is only one shared SNP. F value should be greater than 10 to avoid the weak instrumental variable bias. *p* value < 0.05 was considered for the causal relationships between two traits.

## 4. Discussion

Many potential treatments have been assessed to improve clinical outcomes in sepsis. However, at present, no specific medications for sepsis have been identified [35]. Despite resistin’s integral role in inflammation, little is understood about how human genetics may mediate its impact on the onset or survival of sepsis. Through the examination of single-cell transcriptome data, RETN was found to be upregulated in macrophages during sepsis. Moreover, RETN was also upregulated in sepsis samples compared to healthy controls in bulk transcriptome data. By utilizing MR analyses based on the joint analysis of eQTLs, pQTLs, meQTLs, and GWASs data, a valid test for the presence of a causal relationship between resistin and sepsis can be provided. Our research highlights the particular and protective function of resistin in preventing sepsis-related death.

Resistin was initially discovered in 2001 as a unique signaling molecule secreted by adipocytes and named for its ability to induce insulin resistance [2]. While resistin has been associated with insulin resistance, obesity, and diabetes, it appears to play a more substantial role in inflammation. In a study of patients with acute pancreatitis, resistin levels upon admission were significantly linked to clinical severity and clinical endpoints such as death, serving as an early predictor of peripancreatic necrosis and acute pancreatitis severity [36]. Multiple pieces of evidence also suggest that resistin plays a crucial role in sepsis. For instance, experimental models of human sepsis and studies of critically ill patients with sepsis or septic shock reveal elevated resistin levels. In critically ill patients, resistin is associated with sepsis severity, inflammatory factors, and insulin resistance [37]. In a neonatal sepsis study, resistin levels were higher in septicemic neonates, compared to controls, and those with severe sepsis and those requiring mechanical ventilation, with no significant differences observed between survivors and non-survivors [5]. Furthermore, in our prior research employing single-cell technology, we discovered that RETN was elevated in monocytes or macrophages of patients with sepsis in comparison to healthy controls (unpublished), supporting previous studies [38,39]. In this research, we have disclosed, through robust and causal evidence obtained from MR analyses, the protective role of resistin in sepsis, particularly in critically ill patients.

Resistin is a human body protein that is well tolerated and safe for use in the ICU setting, as our research has shown no causal links between resistin and the failure of organs such as the lung, heart, and kidney. Of particular importance is that we discovered an SNP (rs3745367) that plays a causal role in both RETN protein expression and sepsis pathogenesis through the Mendelian randomization of cis-meQTL, along with a closely related CpG site (cg02346997, cg06633066, and cg22322184). rs3745367 plays a fundamental role in the phosphorylation of RETN, which, in turn, affects resistin protein expression [40]. Our MR and co-location analyses have substantiated this relationship. cg02346997 is situated in the promoter region of RETN and has been found to have a causal relationship with RETN protein expression [41]. We further identified two methylation sites, cg06633066 and cg22322184, that have a relationship with resistin, which provides a basis for the investigation of targets and mechanisms.

This study has a few limitations that need to be acknowledged. First, the analyses were limited to individuals of European ancestry, and further research is necessary to determine the applicability of these results to other ancestries. Second, due to the unavailability of additional outcome data, the survival outcome of sepsis in our study was from a single center, which may limit the generalizability of our findings. Multicenter data could better demonstrate the robustness of our research. Moreover, considering the heterogeneity and complexity of sepsis, more hierarchical data according to the types of infectious agents or primary infection are needed. Third, our evidence is from genetics that reflect the long function of resistin, so more evidence from experiments is warranted. Finally, it is unclear if resistin plays a role in sepsis through some intermediary factor in the pathway. In our study, we attempted to identify such a factor, including CAP1 (cyclase-associated actin cytoskeleton regulatory protein 1, a receptor of resistin), TLR4 (Toll-like receptor 4, a receptor of resistin), and insulin, but we were unable to find any evidence of such a factor. The search for an intermediary factor is challenging with MR analysis and requires knowledge of multiple dimensions, multi-omics, and concurrent evidence from other studies such as molecular experiments. We hope that future studies will provide further insight into our research through other levels of evidence. Despite these limitations, point estimates for the genetically predicted effect of resistin on sepsis death were consistent across multi-QTL data, which is encouraging.

In summary, our study’s MR analyses provide support for a protective effect of resistin in sepsis and indicate potential for drug development. The consistent results across multi-QTL data and GWAS data of sepsis death lend further robustness to our findings. However, the detailed mechanisms underlying this protective effect remain unclear and require further investigation.

## 5. Conclusions

RETN was found to be upregulated in macrophages during sepsis through the analyses of single-cell transcriptome data. Based on MR analyses of multi-omics QTL and GWAS data, we provide evidence for a causal effect of resistin on sepsis death, which is a crucial step towards the development of effective treatments for this life-threatening condition. The identification of two CpG sites associated with sepsis death and RETN protein levels provides additional insight into the mechanisms underlying the protective effect of resistin in sepsis.

## Figures and Tables

**Figure 1 ijms-24-14982-f001:**
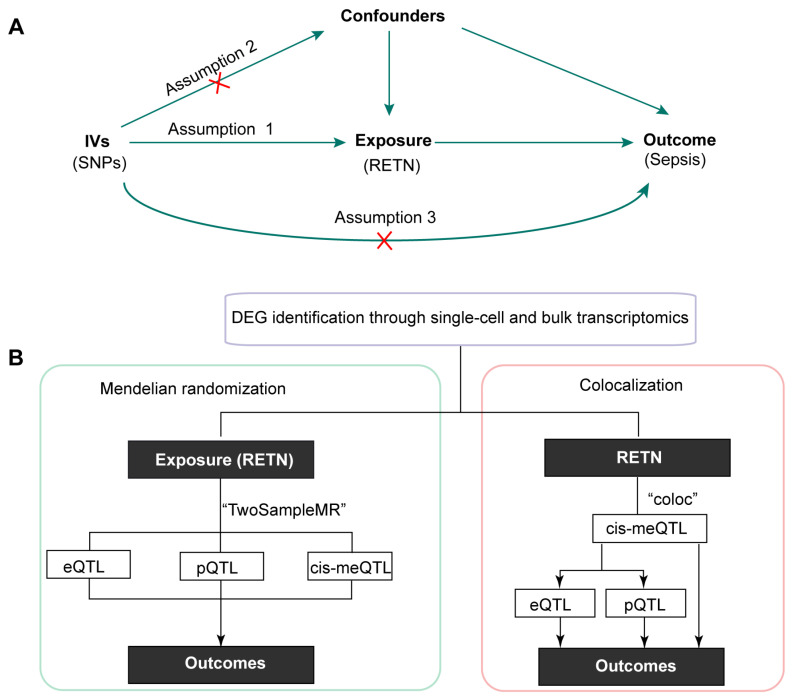
(**A**) The schematic diagram of this study (IVs: instrumental variables; SNPs: single nucleotide polymorphisms). (**B**) The overall design of this research (DEG, differentially expressed genes; QTL: quantitative trait loci; RETN, resistin).

**Figure 2 ijms-24-14982-f002:**
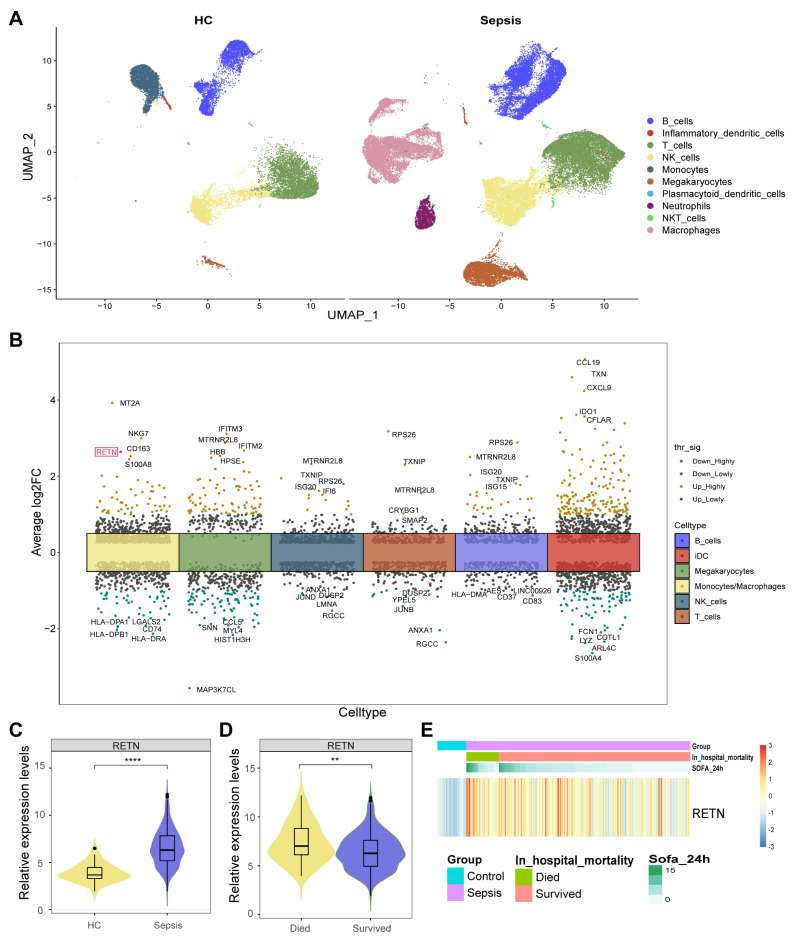
(**A**) All cells were classified into ten types and displayed in two UMAP. (**B**) We identified 560 cell type−specific DEGs in six cell types between healthy controls and sepsis samples. RETN was upregulated in macrophages during sepsis compared with healthy controls and was highlighted in red. thr_sig: Threshold_significance. log2FC: log2 (fold change). (**C**) The violin plot shows the upregulated expression levels of RETN in sepsis samples compared to healthy controls in bulk transcriptome data (****: *p* < 0.0001). (**D**) The violin plot shows the upregulated expression levels of RETN in samples of deceased patients compared to controls of survivors (**: *p* < 0.01). (**E**) The heatmap displays the normalized expression levels of RETN in each sample (SOFA: Sequential Organ Failure Assessment).

**Figure 3 ijms-24-14982-f003:**
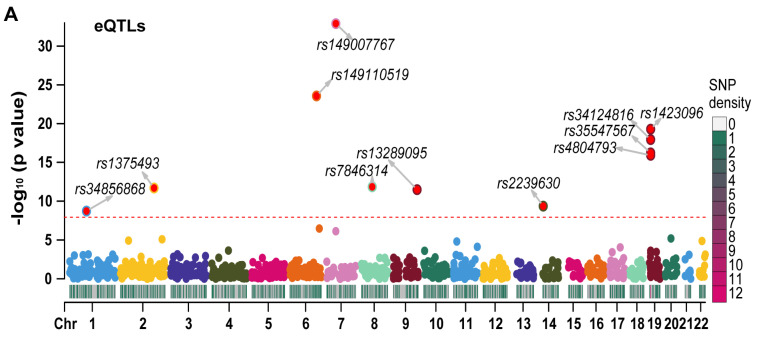
The Manhattan plots for eQTLs and pQTLs of RETN. (**A**) The Manhattan plot for independent eQTLs of RETN (gene symbol of resistin) after LD−clumping (Chr: chromosome, LD: linkage disequilibrium). (**B**) The Manhattan plot for independent pQTLs of RETN after LD−clumping.

**Figure 4 ijms-24-14982-f004:**
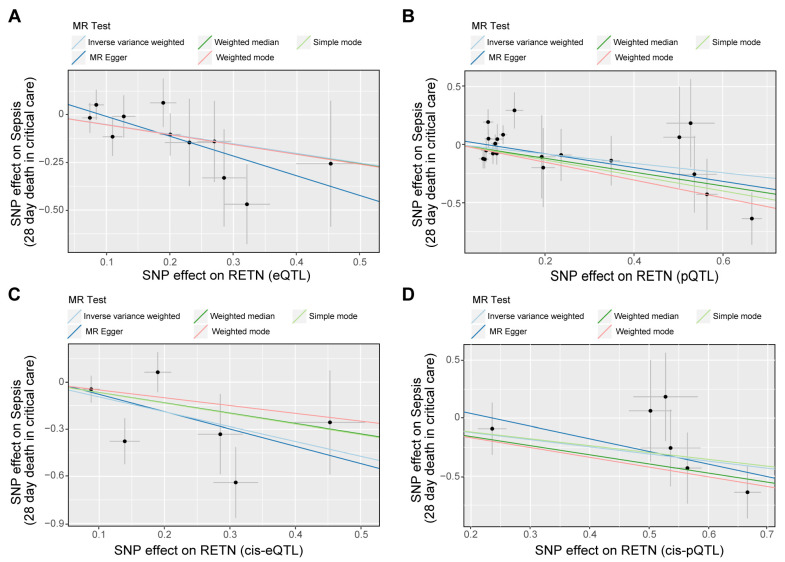
The scatter plots for the associations between exposure and outcome through Mendelian randomization (MR) analyses. (**A**–**D**) The scatter plot shows the negative associations between RETN (eQTLs as IVs (**A**), pQTLs as IVs (**B**), cis-eQTLs as IVs (**C**), and cis-pQTLs as IVs (**D**)) and sepsis death. Five fitted curves indicated different mendelian test methods and inverse variance weighted (IVW) was the main and preferential method.

**Figure 5 ijms-24-14982-f005:**
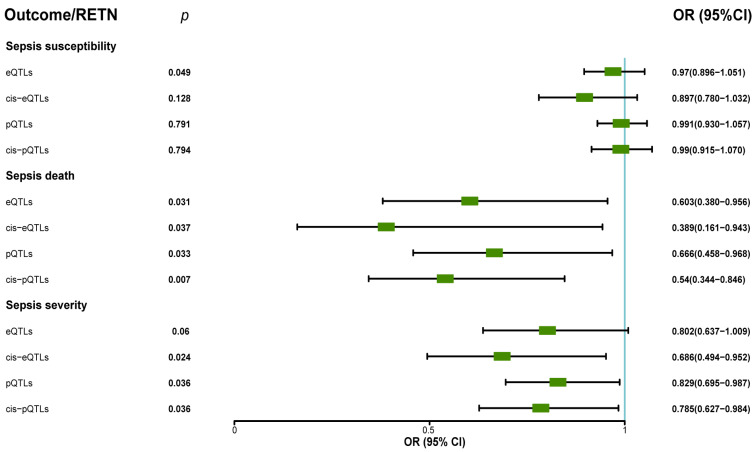
The forest plot displayed the protective roles of exposure (resistin) in sepsis death through MR analyses when we used eQTLs, pQTLs, cis–eQTLs, and cis–pQTLs as IVs (OR: odds ratio; CI: confidence interval). The forest plot also showed the protective roles (OR < 1, *p* < 0.05) of exposure (resistin) in sepsis severity when we used pQTLs, cis–eQTLs, and cis–pQTLs as IVs.

**Table 1 ijms-24-14982-t001:** The *p* value, OR, and 95% CI of resistin levels (eQTLs, cis-eQTLs, pQTLs, and cis-pQTLs) and all outcomes.

IEU ID (Outcome)	Trait	RETN (Exposure)	*p*	OR	95% CI
ieu-b-4980	Sepsis	eQTLs	0.049	0.970	0.896–1.051
		cis-eQTLs	0.128	0.897	0.780–1.032
		pQTLs	0.791	0.991	0.930–1.057
		cis-pQTLs	0.794	0.990	0.915–1.070
ieu-b-4981	Sepsis (28 day death in critical care)	eQTLs	0.031	0.603	0.380–0.956
		cis-eQTLs	0.037	0.389	0.161–0.943
		pQTLs	0.033	0.666	0.458–0.968
		cis-pQTLs	0.007	0.540	0.344–0.846
ieu-b-4982	Sepsis (critical care)	eQTLs	0.060	0.802	0.637–1.009
		cis-eQTLs	0.024	0.686	0.494–0.952
		pQTLs	0.036	0.829	0.695–0.987
		cis-pQTLs	0.036	0.785	0.627–0.984
ieu-b-4979	Pneumonia (death)	eQTLs	0.012	0.754	0.604–0.940
		cis-eQTLs	0.121	0.770	0.554–1.071
		pQTLs	0.292	0.927	0.805–1.067
		cis-pQTLs	0.342	0.871	0.656–1.157
ieu-b-4978	Pneumonia (critical care)	eQTLs	0.205	0.872	0.707–1.077
		cis-eQTLs	0.904	0.975	0.646–1.472
		pQTLs	0.200	1.091	0.955–1.247
		cis-pQTLs	0.488	1.091	0.853–1.394
ieu-b-4977	Pneumonia (28-day death in critical care)	eQTLs	0.658	0.901	0.567–1.430
		cis-eQTLs	0.693	1.182	0.515–2.716
		pQTLs	0.913	0.982	0.712–1.356
		cis-pQTLs	0.491	0.882	0.616–1.262
ebi-a-GCST90000255	Severe COVID-19 infection with respiratory failure (analysis I)	eQTLs	0.697	1.071	0.757–1.516
		cis-eQTLs	0.495	1.273	0.637–2.545
		pQTLs	0.785	1.043	0.769–1.416
		cis-pQTLs	0.508	1.148	0.763–1.727
ebi-a-GCST90000256	Severe COVID-19 infection with respiratory failure (analysis II)	eQTLs	0.851	1.044	0.668–1.630
		cis-eQTLs	0.761	0.900	0.456–1.775
		pQTLs	0.834	0.967	0.703–1.329
		cis-pQTLs	0.683	0.919	0.612–1.379
ukb-d-I9_K_CARDIAC	Death due to cardiac causes	eQTLs	0.558	1.000	0.999–1.002
		cis-eQTLs	0.768	1.000	0.999–1.001
		pQTLs	0.984	1.000	0.999–1.001
		cis-pQTLs	0.399	1.001	0.999–1.002
ebi-a-GCST009541	Heart failure	eQTLs	0.115	0.957	0.907–1.011
		cis-eQTLs	0.135	0.941	0.869–1.019
		pQTLs	0.327	0.975	0.927–1.026
		cis-pQTLs	0.084	0.948	0.891–1.007
ukb-d-I50	Heart failure	eQTLs	0.332	1.000	0.999–1.000
		cis-eQTLs	0.040	0.999	0.998–1.000
		pQTLs	0.576	1.000	1.000–1.000
		cis-pQTLs	0.681	1.000	0.999–1.001
finn-b-N14_RENFAIL	Renal failure	eQTLs	0.621	1.031	0.913–1.164
		cis-eQTLs	0.337	1.100	0.905–1.337
		pQTLs	0.262	1.059	0.958–1.171
		cis-pQTLs	0.609	1.044	0.885–1.232
ukb-b-4963	Acute renal failure	eQTLs	0.573	1.001	0.999–1.003
		cis-eQTLs	0.982	1.000	0.997–1.003
		pQTLs	0.393	0.999	0.998–1.001
		cis-pQTLs	NA ^a^	NA	NA
finn-b-K11_HEPFAIL	Hepatic failure	eQTLs	0.836	0.957	0.631–1.451
		cis-eQTLs	0.509	0.804	0.421–1.536
		pQTLs	0.514	1.122	0.794–1.586
		cis-pQTLs	0.660	1.112	0.693–1.784
finn-b-DEATH	Any death	eQTLs	0.698	0.976	0.864–1.103
		cis-eQTLs	0.153	1.127	0.956–1.328
		pQTLs	0.463	1.032	0.949–1.123
		cis-pQTLs	0.232	1.096	0.943–1.274

^a^ NA indicates there is no independent instrumental variable for this Mendelian randomization analysis.

**Table 2 ijms-24-14982-t002:** The *p* value, OR, and 95% CI of CpG sites directly associated with RETN (cis-meQTLs) and outcomes of sepsis severity and death.

CpG Sites (Exposure)	Cis-meQTL (SNP)	Outcome	*p*	OR	95%CI
cg02346997	rs3745367	RETN (eQTLs)	0.085	0.954	0.904–1.007
		RETN (pQTLs)	1.224 × 10^−42^	0.742	0.711–0.775
		Sepsis death	0.004	1.709	1.183–2.469
		Sepsis severity	0.378	1.086	0.905–1.304
cg02383368	rs4134860	RETN (eQTLs)	0.010	0.859	0.765–0.965
		RETN (pQTLs)	0.224	0.947	0.867–1.034
		Sepsis death	0.252	1.586	0.721–3.489
		Sepsis severity	0.064	1.451	0.978–2.151
cg06633066	rs3745367	RETN (eQTLs)	0.085	0.859	0.722–1.021
		RETN (pQTLs)	1.224 × 10^−42^	0.383	0.334–0.440
		Sepsis death	0.004	5.601	1.717–18.274
		Sepsis severity	0.378	1.303	0.723–2.346
cg11931253	rs72990846 rs8107343	RETN (eQTLs)	0.055	0.851	0.721–1.004
		RETN (pQTLs)	0.740	1.052	0.779–1.422
		Sepsis death	0.963	0.984	0.506–1.914
		Sepsis severity	0.978	1.005	0.719–1.404
cg15460739	rs4134849	RETN (eQTLs)	3.830 × 10^−45^	1.425	1.204–1.686
		RETN (pQTLs)	0.740	1.052	0.779–1.422
		Sepsis death	0.778	0.860	0.302–2.454
		Sepsis severity	0.243	0.731	0.432–1.237
cg15576517	rs34205585 rs807812	RETN (eQTLs)	0.664	1.011	0.961–1.065
		RETN (pQTLs)	0.473	1.039	0.935–1.155
		Sepsis death	0.944	0.980	0.550–1.745
		Sepsis severity	0.761	0.974	0.825–1.151
cg15828235	rs72994460	RETN (eQTLs)	0.007	1.098	1.026–1.174
		RETN (pQTLs)	0.340	1.156	0.859–1.555
		Sepsis death	0.691	0.846	0.372–1.927
		Sepsis severity	0.759	0.938	0.625–1.409
cg17474222	rs10406687	RETN (eQTLs)	0.001	1.303	1.117–1.520
		RETN (pQTLs)	9.350 × 10^−6^	1.337	1.176–1.520
		Sepsis death	0.164	2.145	0.733–6.277
		Sepsis severity	0.990	1.003	0.586–1.717
cg18563630	rs583984 rs794083	RETN (eQTLs)	0.272	0.931	0.820–1.058
		RETN (pQTLs)	NA ^a^	NA	NA
		Sepsis death	0.643	1.122	0.689–1.827
		Sepsis severity	0.932	0.990	0.787–1.246
cg22322184	rs3745367	RETN (eQTLs)	0.085	0.943	0.881–1.008
		RETN (pQTLs)	1.224 × 10^−42^	0.690	0.654–0.728
		Sepsis death	0.004	1.949	1.233–3.081
		Sepsis severity	0.378	1.108	0.882–1.391
cg24433207	rs4134825	RETN (eQTLs)	2.827 × 10^−5^	1.394	1.193–1.628
		RETN (pQTLs)	0.166	1.087	0.966–1.222
		Sepsis death	0.798	0.881	0.334–2.325
		Sepsis severity	0.195	0.724	0.445–1.179
cg24759919	rs147516010 rs794077 rs599330	RETN (eQTLs)	0.209	1.038	0.979–1.100
		RETN (pQTLs)	0.875	1.009	0.898–1.135
		Sepsis death	0.873	1.028	0.735–1.436
		Sepsis severity	0.832	1.018	0.862–1.203

^a^ NA indicates there is no independent instrumental variable for this Mendelian randomization analysis.

**Table 3 ijms-24-14982-t003:** The colocalization results of eQTLs, pQTLs of RETN, and cis-meQTLs within RETN gene or promoter region, with the main traits of sepsis.

Trait1	Trait2	PP.H1	PP.H2	PP.H3	PP.H4
RETN-eQTL	Sepsis death	0.487	0.000	0.437	0.075
RETN-pQTL		0.159	0.000	0.153	0.688
Cis-meQTL		0.455	0.000	0.000	0.545
RETN-eQTL	Sepsis severity	0.557	0.000	0.363	0.080
RETN-pQTL		0.158	0.000	0.143	0.700
Cis-meQTL		0.968	0.000	0.000	0.032
Cis-meQTL	RETN-eQTL	0.000	0.000	1.000	0.000
	RETN-pQTL	0.000	0.000	0.000	1.000

## Data Availability

Single-cell transcriptome data are from GEO database (https://www.ncbi.nlm.nih.gov/geo/query/acc.cgi?acc=GSE167363, accessed on 15 December 2022). Bulk transcriptome data are from GEO database (https://www.ncbi.nlm.nih.gov/geo/query/acc.cgi?acc=GSE185263, accessed on 15 December 2022). The GWAS data are available in IEU Open GWAS Project (https://gwas.mrcieu.ac.uk/, accessed on 10 February 2023). The eQTL data of RETN can be found in the eQTLGen Consortium (https://eqtlgen.org/index.html, accessed on 10 February 2023) or IEU Open GWAS Project. The pQTL data of RETN are available in deCODE dataset (https://www.decode.com/summarydata/, accessed on 22 February 2023). The meQTL data of RETN are available in GoDMC dataset (http://mqtldb.godmc.org.uk/downloads, accessed on 27 February 2023). LD reference data for the European super population can be downloaded directly through the following link (http://fileserve.mrcieu.ac.uk/ld/1kg.v3.tgz, accessed on 15 February 2023).

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
