# Peer review of "Integrated Analyses of Single-Cell Transcriptome and Mendelian Randomization Reveal the Protective Role of Resistin in Sepsis Survival in Intensive Care Unit"

_ijms, 2023, doi:10.3390/ijms241914982_

Round 1
Reviewer 1 Report
The manuscript submitted by Authors deals with the integrated analyses of single-cell transcriptome and Mendelian randomization reveal the protective role of resistin in sepsis survival in intensive care unit. Overall, the manuscript is really interesting. Congratulations to the authors of the topic and interesting results. The paper contains a huge amount of data. Nevertheless, do the authors believe that such conclusions can be drawn on the basis of single-cell transcriptome data from GEO database which contains PBMC only from two healthy donors and five sepsis patients? Can the authors include more transcriptome data in the analysis to make the conclusions more reliable?
Author Response
The manuscript submitted by Authors deals with the integrated analyses of single-cell transcriptome and Mendelian randomization reveal the protective role of resistin in sepsis survival in intensive care unit. Overall, the manuscript is really interesting. Congratulations to the authors of the topic and interesting results. The paper contains a huge amount of data. Nevertheless, do the authors believe that such conclusions can be drawn on the basis of single-cell transcriptome data from GEO database which contains PBMC only from two healthy donors and five sepsis patients? Can the authors include more transcriptome data in the analysis to make the conclusions more reliable?
Response: We totally understand your concerns about the the sample size of the single-cell data and we have to apologize for that. First, when we began to conduct our analyses, there are few single-cell data of sepsis patients and healthy controls available. Even so, we believe the accuracy of the data. For instance, Neutrophils and megakaryocytes proliferated in sepsis in the single-cell data, and MHC-2 related genes are down-regulated in monocytes/macrophages, which are consistent with previous studies and scientific consensus.
Second, thank you for your suggestion about including more transcriptome data in the analysis. We collected bulk transcriptome data from GEO database (GSE185263, https://www.ncbi.nlm.nih.gov/geo/query/acc.cgi?acc=GSE185263) which contains 348 sepsis patients and 44 healthy controls. The transcriptome data processing flow can be found in our revised manuscript. Overall, RETN also was upregulated in sepsis samples compared with healthy controls (Figure 2C, E). Moreover, RETN also was upregulated in died samples compared with survived patients (Figure 2D, E). Thanks again for your precious suggestions which help us extend our research.
Reviewer 2 Report
In this well conducted study the authors arrived at the conclusion that resistin may have a role in sepsis.
It is well known that sepsis is a very complex disease in which almost all immunocytes and others cells of the endothelium, heart, renal, liver and other organs as well have a role. The changes in the concentrations or levels of various proteins, cytokines ad other mediators are altered and this alteration depends on the stage of the disease and the time of collection of the samples for analysis.
So in such a complex disease of sepsis, measuring gene expressions, proteins and other mediators may be very difficult if not impossible. Any sch efforts to measure potential markers of sepsis depends on the selection of the stage of the disease, insult that is responsible for the onset of sepsis, the response of the tissues, organs and the specific patient.
In the present study the finding that resistin has a ole in sepsis is probably applicable to the specific situation but can not be generalised. This limitation of the study need to be emphasized by the authors.
I suggest that authors collect blood samples from different types of sepsis and in different stages of sepsis and measure plasma levels of resistin and add those results to the current work.
ok
Author Response
In this well conducted study the authors arrived at the conclusion that resistin may have a role in sepsis.
It is well known that sepsis is a very complex disease in which almost all immunocytes and others cells of the endothelium, heart, renal, liver and other organs as well have a role. The changes in the concentrations or levels of various proteins, cytokines ad other mediators are altered and this alteration depends on the stage of the disease and the time of collection of the samples for analysis.
So in such a complex disease of sepsis, measuring gene expressions, proteins and other mediators may be very difficult if not impossible. Any sch efforts to measure potential markers of sepsis depends on the selection of the stage of the disease, insult that is responsible for the onset of sepsis, the response of the tissues, organs and the specific patient.
In the present study the finding that resistin has a ole in sepsis is probably applicable to the specific situation but can not be generalised. This limitation of the study need to be emphasized by the authors.
I suggest that authors collect blood samples from different types of sepsis and in different stages of sepsis and measure plasma levels of resistin and add those results to the current work.
Response: Those you mentioned are interesting questions and we totally understand your concerns. First, sepsis is a complex disease and potential markers identification is a difficult task. And that is why we Looked for evidence in a variety of data including single-cell, eQTL, pQTL, meQTLand GWAS. So, we believe our conclusion is reliable enough.
Second, as your suggestion about collecting blood samples from different types of sepsis and in different stages of sepsis, we did collect bulk transcriptome data from GEO database (GSE185263, https://www.ncbi.nlm.nih.gov/geo/query/acc.cgi?acc=GSE185263) which contains 348 sepsis patients and 44 healthy controls. The transcriptome data processing flow can be found in our revised manuscript. Overall, RETN also was upregulated in sepsis samples compared with healthy controls (Figure 2C, E). Moreover, RETN also was upregulated in died samples compared with survived patients (Figure 2D, E). The SOFA scores of all samples are also displayed in Figure 2E, which indicated that all samples are in multiple stages of sepsis. More research about resistin are in process in our lab and we can not release these results for confidentiality reasons at this time. Thanks again for your precious suggestions which help us extend our research.
Round 2
Reviewer 1 Report
I have no further comments.
Author Response
Thanks again for your precious suggestions again.
Reviewer 2 Report
Sepsis is a complex disease. So looking at resistin alone is not sufficient. Authors need to look at other molecules such as insulin, adiponectin, TNF, IL-6, GLP-1, GLP-2, ghrelin, leptin, etc., and arrive at a comprehensive analysis of sepsis pathobiology.
It has been shown that GIK regimen may be of benefit in sepsis and other inflammatory conditions. There are many lipids that regulate insulin resistance and ay have a role in sepsis.
It is also possible that the role of resistin noted in the current study may be the result of sepsis instead of cause of sepsis. Thus, the role of resistin in sepsis need to be evaluated whether it is the cause or effect of the sepsis process.
In a clinical situation the authors need to look at the plasma levels of resistin, insulin, adiponectin, ghrelin, leptin and the effect of effect of GIK regimen and consequent change sin GLP and other factors including IL-6,TNF , and HMGB1 in sepsis and correlate the same to its prognosis and outcome.
ok
Author Response
Sepsis is a complex disease. So looking at resistin alone is not sufficient. Authors need to look at other molecules such as insulin, adiponectin, TNF, IL-6, GLP-1, GLP-2, ghrelin, leptin, etc., and arrive at a comprehensive analysis of sepsis pathobiology.
It has been shown that GIK regimen may be of benefit in sepsis and other inflammatory conditions. There are many lipids that regulate insulin resistance and ay have a role in sepsis.
It is also possible that the role of resistin noted in the current study may be the result of sepsis instead of cause of sepsis. Thus, the role of resistin in sepsis need to be evaluated whether it is the cause or effect of the sepsis process.
In a clinical situation the authors need to look at the plasma levels of resistin, insulin, adiponectin, ghrelin, leptin and the effect of effect of GIK regimen and consequent change sin GLP and other factors including IL-6, TNF, and HMGB1 in sepsis and correlate the same to its prognosis and outcome.
Response: Those you mentioned are interesting questions and we admire your extensive knowledge about sepsis. Sepsis is a complex disease and potential markers identification is a difficult task. Even we looked for evidence from a variety of data including single-cell and bulk transcriptome, eQTL, pQTL, meQTLand GWAS, we could not address all your concerns.
First, the molecules you mentioned are important in sepsis research, however, the first part of our research focused on the peripheral blood transcriptome of sepsis patients and healthy controls. And it is well-known that Islet cells in the pancreas are responsible for producing insulin and glucagon, and leptin and adiponectin are secreted by adipocytes. So, these molecules could not be detected in the peripheral blood transcriptome. Even so, as your suggestions, we did explore the expression levels of some molecules including IL6, TNF, HMGB1, GLP1R, LEPR, ADIPOR1, ADIPOR2 in the bulk transcriptome.
Second, resistin could be secreted by monocytes, macrophages and neutrophils and are elevated in sepsis. Resistin is the molecule we want to focus on. Even though more clinical research is needed, we did prove the robustness our results from multi omics data and studies.
Third, as you mentioned that resistin may be the result of sepsis instead of cause of sepsis. We also noticed that possibility and we could not agree more that transcriptome analysis can only observe phenomena instead of associations. So, that is the question we provided our solution using genetics and mendelian randomization (MR) method. By using the MR paradigm, our findings are less vulnerable to the environmental confoundings and reverse causation bias that can hinder causal inference in traditional epidemiological study designs. Additionally, by incorporating genetic association summary data we use human-centric analysis with more directly translatable findings, thereby bypassing the key limitation of animal models. Finally, the use of publicly available genetic association summary data makes our approach maximally time and cost-efficient.
Finally, more research about resistin are in process in our lab and we did find that resistin could be secreted by monocytes and macrophages and are elevated in sepsis. These results are unpublished and we can not release these results for confidentiality reasons at this time. Thanks again for your precious suggestions which help us extend our research. We apologize again that we could not answer all of your questions in this research.